# A Simple and Convenient Method for Preparing Fluorine-Free Durable Superhydrophobic Coatings Suitable for Multiple Substrates

**DOI:** 10.3390/ma16051771

**Published:** 2023-02-21

**Authors:** Bin Xu, Yinping Zhou, Shichang Gan, Qinqin Xu, Maohua Hou, Congda Lu, Zhongjin Ni

**Affiliations:** 1College of Materials Science and Engineering, Zhejiang University of Technology, Hangzhou 310014, China; 2College of Mechanical Engineering, Zhejiang University of Technology, Hangzhou 310014, China; 3College of Engineering, Zhejiang Agriculture and Forestry University, Hangzhou 311300, China

**Keywords:** superhydrophobic coatings, SiO_2_, SBS, C9 petroleum resin, spraying

## Abstract

Superhydrophobic coatings have attracted a lot of attention due to their excellent self-cleaning and anti-fouling capabilities. However, the preparation processes for several superhydrophobic coatings are intricate and expensive, which restricts their usefulness. In this work, we present a straightforward technique for creating durable superhydrophobic coatings that can be applied to a variety of substrates. The addition of C9 petroleum resin to a styrene-butadiene-styrene (SBS) solution lengthens the SBS backbone and undergoes a cross-linking reaction to form a dense spatial cross-linked structure, improving the storage stability, viscosity, and aging resistance of the SBS. The combined solution functions as a more stable and effective adhesive. Using a two-step spraying technique, the hydrophobic silica (SiO_2_) nanoparticles solution was applied to the surface to create durable nano-superhydrophobic coatings. Additionally, the coatings have excellent mechanical, chemical, and self-cleaning stability. Furthermore, the coatings have wide application prospects in the fields of water–oil separation and corrosion prevention.

## 1. Introduction

In recent years, scientists have increasingly researched biological surfaces, including lotus leaves [1], waterfly legs [2], cicada wings [3], and other surfaces with high dust-proofing and self-cleaning functions. A large number of studies on superhydrophobic surfaces using biological surfaces as templates have revealed that a rough structure and low surface energy are necessary conditions for superhydrophobic surfaces [4,5,6]. The contact angle (CA) of the superhydrophobic surface is greater than 150°, and the sliding angle (SA) is less than 10°. Superhydrophobic surfaces have been widely used in self-cleaning [7], anti-icing [8], anti-corrosion [9,10], drag reduction [11], oil–water separation [12,13], and other fields. However, superhydrophobic surfaces have poor stability, and the surface is destroyed easily in special environments, such as mechanical friction or chemical corrosion. Additionally, most superhydrophobic materials have limitations such as high costs, complex fabrication processes, and a single substrate. As a result, strong superhydrophobic coatings that are low-cost, easy to apply, and can be employed on a variety of substrates are required.

At present, superhydrophobic coatings are made using a variety of techniques, including the sol–gel approach [14,15], template method [16,17], plasma etching method [18,19,20], chemical vapor deposition [21,22], self-assembly [23], rapid breakdown anodization [24], chemical etching [25], and others. However, the substrate is severely constrained by these procedures, because these methods of making superhydrophobic coatings are complicated and expensive. In contrast, the wet chemical processes of dipping and spraying are of particular interest, and spraying in particular is desirable. Because the spraying method is more convenient for large-scale manufacturing and less dependent on equipment, there is no requirement for the size and shape of the substrate [26,27]. Some researchers prepared superhydrophobic coatings by mixing nanoparticles directly with adhesives with low surface energy [28,29,30,31,32]. The addition of nanoparticles provides a rough structure, which is an important factor in the superhydrophobic coating [33]. In early studies, silica and titanium dioxide as nanoparticles, and epoxy resin, SBS, and polydimethylsiloxane (PDMS) as common low surface energy adhesives have been widely studied in the preparation of superhydrophobic coatings. However, large amounts of nanoparticles are added, which makes the durability of the coating poor. Some other scholars have explored a strategy similar to “spraying adhesive + functional coating” to enhance the adhesion between coating and substrate, and to improve the mechanical wear resistance and durability of the surface [34,35]. First, a layer of primer resin coating is sprayed to improve the durability and stability of the coating, and then a layer of nanoparticles with low surface energy is sprayed on the surface to construct a superhydrophobic surface with roughness.

Recently, SBS was widely utilized in coatings [36], adhesives [37], inks [29,38], and other materials because of its high tensile strength, superior chemical resistance, outstanding low-temperature characteristics, and low cost. SBS has been widely employed to create superhydrophobic coatings in recent years [29,30,31,32]. Some scholars prepared superhydrophobic coatings by mixing SBS and nanoparticles with low surface energy directly in an organic solvent. Since SBS is non-polar and has a weak van der Waals force and permeability, the bonding effect is not immediately apparent. It can have a stronger bonding effect if petroleum resin is used with it [39,40]. The storage stability, viscosity, and anti-aging of SBS have been reported to be improved by adding petroleum resin to the SBS solution, which lengthens and cross-links the SBS skeleton to produce a dense spatial cross-linked structure [36,40,41].

Based on previous work, we prepared robust superhydrophobic coatings by mixing C9 petroleum into SBS solution, spraying the resulting mixture as a binder layer on a glass substrate while it was still outside, and then covering the surface with a hydrophobic SiO_2_ nanoparticle solution. The mass ratio of SBS to C9 petroleum resin was 1:2, and the hydrophobic SiO_2_ nanoparticles content was 5 wt.% to produce the SBS, C9 petroleum resin and hydrophobic SiO_2_ nanoparticles (SBS/C9-SiO_2_) coatings with the optimum performance. The CA was 155.6° ± 1° and the SA was 4.1° ± 0.5° when SBS/C9-SiO_2_ coatings were put on the glass surface. Spraying the coatings onto a variety of different substrates, such as paper, press cloth, metal, and so on, also rendered it superhydrophobic. The SBS/C9-SiO_2_ coatings have corrosion resistance and can be used in oil–water separation equipment. The method did not need expensive or sophisticated equipment and had a wide range of possible applications.

## 2. Materials and Methods

### 2.1. Materials

Linear SBS (YH-792) was purchased from Dongguan Jishun Rubber & Plastic Technology Co., Ltd. (Dongguan, China). C9 petroleum resin from Shenzhen Jitian Chemical Co., Ltd. (Shenzhen, China). SiO_2_ nanoparticles (SK-200s, 20 nm) was supplied by NANO Technology Co., Ltd (Shaoxing, China). Ethyl acetate was obtained from Ron’s reagent, acetone and ethanol were purchased from Sinopharm, and the glass slides were the Sailboat brand. The filter cloth and paper were purchased from Shaoxing Aobang Technology Co., Ltd. (Shaoxing, China). The aluminum sheet was from Shenzhen Quanfu Metal Co., Ltd. (Shenzhen, China). All chemicals were used as received.

### 2.2. Preparation of the Coatings

The glass slides were sonicated with ethanol and acetone for 10 min to remove surface impurities and oil stains, rinsed with deionized water, and then put into a drying oven at 80 °C for 30 min. SBS and C9 petroleum resin were dissolved in ethyl acetate and stirred at room temperature for 12 h to obtain 9 wt.% SBS and C9 petroleum resin (SBS/C9) solution, and SiO_2_ nanoparticles were then dissolved in ethyl acetate and sonicated for 15 min to produce 5 wt.% of SiO_2_ hydrophobic dispersion. As shown in Figure 1, the SBS/C9 solution was first sprayed on the substrate with the high-pressure (0.2 Mpa) spray gun, kept at room temperature for 1–2 min to wait for the surface to dry, and then a layer of SiO_2_ hydrophobic dispersion was sprayed with a high-pressure (0.2 Mpa) gun and left to dry at room temperature for 5 min to obtain solid SBS/C9-SiO_2_ coatings. The SBS/C9-SiO_2_ coatings could also be prepared by spraying the compound solution on the aluminum sheets, filter cloth, and paper in the same way.

### 2.3. Characterization

The surface micromorphology of the coatings was analyzed by a ΣIGMA 300 field emission scanning electron microscope from Zeiss, Jena, Germany. Before testing, a thin layer of gold was sprayed on the sample surface to increase its electrical conductivity. The functional groups of SBS and C9 petroleum resins were analyzed using the KBr liquid film method. Spectra were recorded on films obtained from solutions after solvent evaporation, with a Fourier transform infrared spectrometer (Nexus, Thermo, Waltham, MA, USA) in the range of 400~4000 cm^−1^. The CA and SA of SBS/C9-SiO_2_ coatings were investigated using a fully automated contact angle meter OCA30 at 8 µL per drop. For each sample, three places were chosen to measure the CA and SA, and the final average value was used to determine the CA and SA of the sample. An electrochemical workstation (CHI 660, Shanghai) was used to test the corrosion resistance of SBS/C9-SiO_2_ coatings. In a 3.5 wt.% NaCl electrolyte solution, saturated potassium chloride was used as the reference electrode, a platinum electrode (Pt) served as the counter electrode, and an exposed 3.14 cm^2^ sample served as the working electrode. The kinetic potential polarization curves of substrates coated with SBS/C9-SiO_2_ on 316 stainless steel and Al were measured. The mechanical stability of the coatings was tested by friction with sandpaper. The glass coated with SBS/C9-SiO_2_ was immersed in three different solutions of sodium chloride solution (3 wt.%), hydrochloric acid solution (pH = 1), and ammonia solution (pH = 13) for 24 h, respectively, in order to test the chemical stability of the coatings.

## 3. Results

### 3.1. Tape Peel Cycle Test

The content of SBS in the adhesive layer remained constant at 3 wt.%, whereas the content of C9 petroleum resin grew gradually from 0 to 8 wt.%, and SiO_2_ nanoparticles were dissolved in ethyl acetate to produce 5 wt.% of SiO_2_ hydrophobic dispersion. When the C9 petroleum resin content is 0, the coatings are recorded as SBS-SiO_2_. The SBS/C9-SiO_2_ coatings with various C9 petroleum resin concentrations were applied to the glass surface by the two-step spraying method. We counted the instances in which the SBS/C9-SiO_2_ coatings could be peeled off the tape after it had dried, or instances in which the coatings either lost their superhydrophobic properties or their CA dropped below 150°. The tape may be peeled off more times thanks to C9 petroleum resin, as shown in Figure 2. This is because C9 petroleum resin increased the number of times the tape peel cycles the SBS/C9-SiO_2_ coatings. The addition of C9 petroleum resin makes the skeleton of SBS longer, and the cross-linking reaction occurs to form a dense spatial cross-linking structure, which improves the storage stability, viscosity, and anti-aging properties of SBS/C9 coatings [42]. When the addition of C9 petroleum resin was 6 wt.%, the coatings had the best mechanical properties and could be peeled by the tape up to 65 times.

### 3.2. Superhydrophobic Coatings Applied to a Variety of Substrates

Dataphysics’s OCA30 automatic contact angle measuring instrument was used to measure the CA. Due to the low surface tension of the SBS/C9-SiO_2_ coatings and the 4 μL water droplet having light gravity, it was difficult to adhere to the surface of the coating. As shown in Figure 3a, a 4 μL drop of water was squeezed out of the dropper of the contact angle meter and slowly lowered to touch and squeeze to the surface of SBS/C9-SiO_2_ coatings, and then lifted the dropper to separate the drop from the surface. Although the water droplets were squeezed and deformed on the surface of the coating, there was no liquid residue on the surface of the coatings after separation, and the droplets recovered to their original state. The above results demonstrated that the coatings had low adhesion to water. As shown in Figure 3b, we employed 8 μL of water droplets and let them fall on the surface of the SBS/C9-SiO_2_ coatings by gravity, then tested the CA of the coatings on aluminum sheets, filter cloth, paper, and glass surfaces, and found that all their CAs were larger than 150°. The SA of the coatings on the substrates was tested using 10 μL of water droplets, and the SAs were all less than 10°, indicating that they all obtained sensational superhydrophobicity.

### 3.3. Surface Wettability and Morphology

Figure 4 shows the microscopic morphology and WCA images of the clean glass, SBS/C9 coatings, and SBS/C9-SiO_2_ coatings’ surface. As demonstrated in Figure 4a, the surface of the untreated clean glass was highly smooth, with a CA of 44.3° ± 1°. As illustrated in Figure 4b, the surface with the SBS/C9 coatings displayed some irregular micro-scale protrusions, and the WCA was 81.7° ± 1° while still maintaining a hydrophilic state. The surface structure following the SBS/C9-SiO_2_ coatings is shown in Figure 4c–e, indicating a multi-scale rough structure, and the surface had a lotus-like surface papillary structure [43] that was formed by the aggregation of the nanoparticles’ uneven aggregates. Figure 4e indicates a surface with many pores of different sizes, which might be explained by the “respiration diagram” [44], which was caused by the high relative humidity of the air when the SBS/C9-SiO_2_ coatings were dry in the air. At this time, the CA of the SBS/C9-SiO_2_ coatings was 155.6° ± 1°, showing a superhydrophobic state, which was mainly determined by the micro-scale pores, SiO_2_ nanoparticle aggregation, and the multi-scale roughness of SiO_2_ nanoparticles.

### 3.4. Surface Composition Analysis

As shown in Figure 5, in the functional group region (4000~1300 cm^−1^), the absorption peaks at 3000~2800 cm^−1^ in the spectra of the SBS/C9 mixture represent the bending vibration of symmetric methylene on the SBS skeleton and the stretching vibration of a C-H bond saturated with C9 petroleum resin. The absorption peak at 1738 cm^−1^ is characteristic of the carbonyl group, due to the presence of ethyl acetate. The peak at 1600 cm^−1^ is the respiratory vibration peak of the benzene skeleton. The peaks at 1490 cm^−1^ and 1448 cm^−1^ are the flexion vibrations of saturated C-H bonds of the SBS and C9 petroleum resins. In the fingerprint region (1300~400 cm^−1^), the peak at 972 cm^−1^ may be offset by the peak of the SBS and C9 petroleum resins at around 1070 cm^−1^, which may be due to the existence of van der Waals forces between them. The peak at 969 cm^−1^ is the absorption peak of the butadiene trans vibration of the SBS. The peak at 910 cm^−1^ is the in-plane vibration of the SBS trans double bond hydrocarbon. The peaks at 749 cm^−1^ and 700 cm^−1^ are the out-of-plane bending vibrations of the benzene C-H bond of the SBS and C9 petroleum resins. There was no obvious change in the mixture spectra, which indicated that the SBS and C9 petroleum resins were physically doped [45].

### 3.5. Mechanical Performance Test

Figure 6a shows a schematic diagram of the sandpaper abrasion test. The sample was applied uniformly onto 800 mesh sandpaper after a 100 g weight was placed on the glass. The distance between each abrasion movement was 20 cm, and the CA of the sample was measured after 10 abrasions, as shown in Figure 6b. As shown in Figure 6c, the SBS-SiO_2_ coatings were badly worn after 50 rubbings, with a CA of 142.8° ± 3°. The severely worn areas exposed the glass substrate, and water droplets did not roll off the surface easily, even though the water contact angle hysteresis was severe. As shown in Figure 7d, there was no significant change in the wettability of the SBS/C9-SiO_2_ coatings after 65 abrasions, with a CA of 149.8° ± 1°. The droplets rolled off the surface easily, despite some of the SiO_2_ nanoparticles having worn off. The reason for this is that the SBS/C9 binder is coated with a layer of nanoparticles during the curing process, which enhances the bond strength between the nanoparticles and the substrate. In addition, the SBS/C9 liquid binder forms micron-sized protrusions when bent, which protect the nanoparticles on the sidewalls and between the protrusions during the wear process, thus improving the mechanical durability of the surface.

### 3.6. A Self-Cleaning Performance of the Superhydrophobic Coatings

As shown in Figure 7, the slide coated with superhydrophobic coatings was placed at an angle. To facilitate observation, we placed the soil on the surface of the glass slide, and then water drops were dropped from the top end of the samples with a dropper. As the water drops rolled down, the soil on the surface rolled with them, and the surface became very clean after a while. On the one hand, the low adhesion of the superhydrophobic coatings to water droplets made it easy for water droplets to roll the across surface and difficult for them to stay on there. On the other hand, since the affinity between the soil and water droplets was larger than the affinity between soil and the surface of the superhydrophobic coatings, the soil also rolls off the surface when water droplets do. This demonstrates that the superhydrophobic coatings developed in this study have strong self-cleaning properties and may be employed in real-world applications.

### 3.7. Chemical Stability Test of Superhydrophobic Coatings

As illustrated in Figure 8a, the sprayed glass with the SBS/C9-SiO_2_ coatings was immersed in three different solutions of sodium chloride solution (3 wt.%), hydrochloric acid solution (pH = 1), and ammonia solution (pH = 13) for 24 h. Subsequently, we discovered that the surface of the coatings was not damaged, that it still had a satisfactory superhydrophobic effect, and that the CA was more than 150°, as shown in Figure 8b. In Figure 8a, the SBS/C9-SiO_2_ coatings of the sample were bright silver in solution because the micro-nano structured coatings were hydrophobic and would not be in direct contact with the solution. A large amount of air was stored between the coatings and the solution, forming an air barrier that effectively prevents the irritating solution from directly touching the coatings, thus protecting the coatings from corrosion. This indicates that the coatings are resistant to corrosion and have promise for real-world applications in marine environments.

### 3.8. Anti-Corrosion Performance of the Superhydrophobic Coatings

Corrosion resistance is frequently studied using a potentiodynamic polarization curve. The corrosion voltage corresponds to the tangent of the cathodic and anodic polarization curves, and the corrosion current density corresponds to the corrosion current density. The more negative the corrosion voltage, the greater the corrosion current, and the more likely the corrosion will occur [46]. Figure 9a,b display the potentiodynamic polarization curves of aluminum and 316 stainless steel with the SBS/C9-SiO_2_ coatings’ surfaces in 3.5 wt.% NaCl aqueous solution, respectively. After superhydrophobic treatment, the corrosion voltage of 316 stainless steel rose from −1.023 V to −0.296 V, while the corrosion current density decreased from 3.844 × 10^−6^ A/cm^2^ to 3.272 × 10^−7^ A/cm^2^. The corrosion voltage of Al increased from −1.334 V to −1.057 V, while the corrosion current density declined from 1.211 × 10^−6^ A/cm^2^ to 4.314 × 10^−7^ A/cm^2^. The results of the potentiodynamic polarization tests reveal that the corrosion current density (Icorr) of all coated samples reduced sharply compared to the bare substrates. Meanwhile, the corrosion potential (Ecorr) of all coated samples moved in a more positive direction as compared to that of the substrate, which indicates that the corrosion trend has dramatically improved and the coating effectively provides a protective way for the substrate. The air layer that forms between the SBS/C9-SiO_2_ coating’s superhydrophobicity and the NaCl solution and the resin binder causes it to generate a two-stage protective layer on the surface of multi-scale structures. Among them, the air layer effectively shields the substrate’s surface from corrosion products. The air layer and resin binder form a double layer of protection and improve the substrate’s ability to resist corrosion, particularly in the case of metal materials.

### 3.9. Oil–Water Separation of Superhydrophobic Coatings

The feasibility of the SBS/C9-SiO_2_ coatings for oil and water separation was studied in this experiment. Because the SBS/C9-SiO_2_ coatings are superhydrophobic and superoilphilic, the super hydrophobic filter clothes cannot be penetrated by water, yet oil with low surface energy may readily pass through. As a result, coated filter clothes are seen to be a promising material in the field of water–oil separation. The oil phase was n-hexane, and the aqueous phase was deionized water and stained with methylene blue for observation. Because n-hexane is significantly less thick than water, it floats to the surface when mixed with it. The filter cloth with the SBS/C9-SiO_2_ coatings was put between the top and lower glass tubes. The separation of a mixture of hexane and water is depicted in Figure 10. The SBS/C9-SiO_2_ coatings on the filter cloth allowed n-hexane to pass through and collect in a beaker below, but water stayed in the glass tube in front of the filter cloth. The result reveals that the filter cloth with the SBS/C9-SiO_2_ coatings can separate oil and water combinations.

## 4. Conclusions

By adding C9 petroleum resin to SBS as a binder and applying SiO_2_ to its surface, it was possible to create a stable superhydrophobic coating with an AC of over 150° and an SA with less than 10°. The optimum mechanical properties of the superhydrophobic coatings were achieved when the addition of C9 resin concentration was 6 wt.%. Superhydrophobic coatings that were applied to a variety of substrates, regardless of the substrate surfaces and geometry, were made using a two-step spraying technique. Additionally, SBS/C9-SiO_2_ coatings exhibit outstanding mechanical properties, excellent self-cleaning properties, and corrosion resistance. Moreover, there are numerous application possibilities for SBS/C9-SiO_2_ coatings, which can be prepared in a straightforward, inexpensive manner and utilized for corrosion resistance, oil and water separation, and other purposes.

## Figures and Tables

**Figure 1 materials-16-01771-f001:**
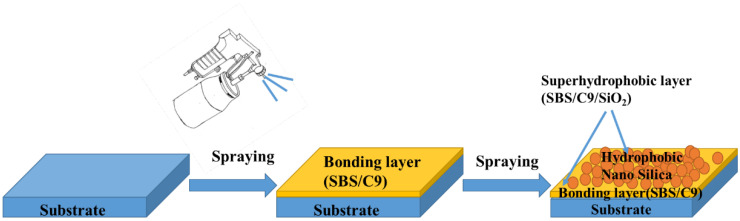
Schematic diagram of the preparation of SBS/C9-SiO_2_ coatings on the substrate.

**Figure 2 materials-16-01771-f002:**
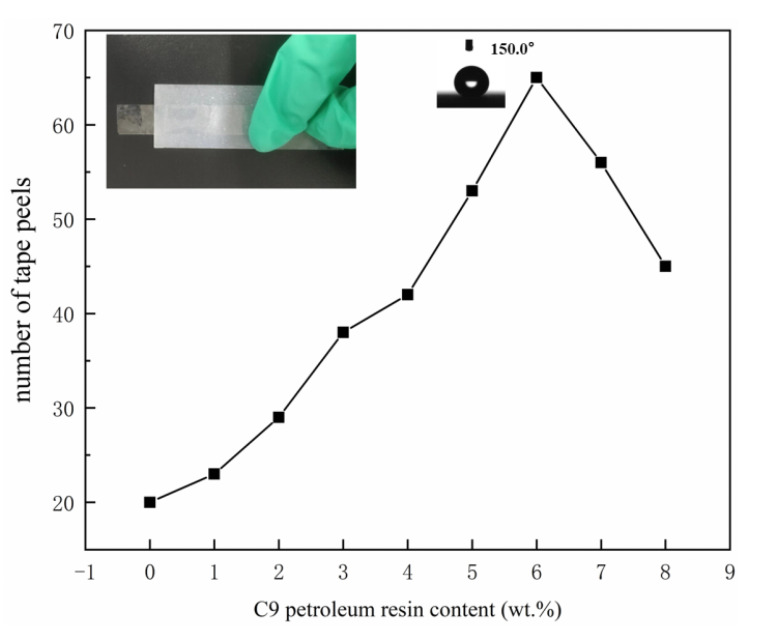
Relationship between the content of C9 petroleum resin and the number of stripping cycles of the SBS/C9-SiO_2_ coating.

**Figure 3 materials-16-01771-f003:**
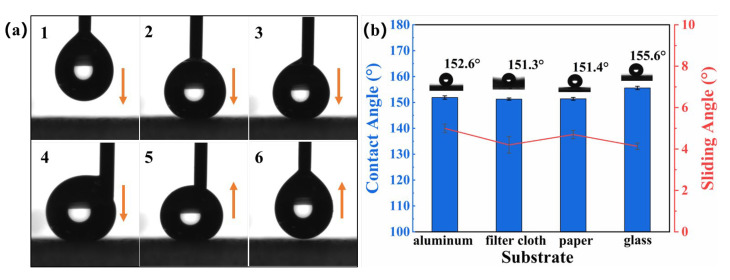
(**a**) Picture of water droplets squeezed and re-lifted on the SBS/C9-SiO_2_ coatings’ surface. (**b**) CA and SA of SBS/C9-SiO_2_ coatings on different substrates.

**Figure 4 materials-16-01771-f004:**
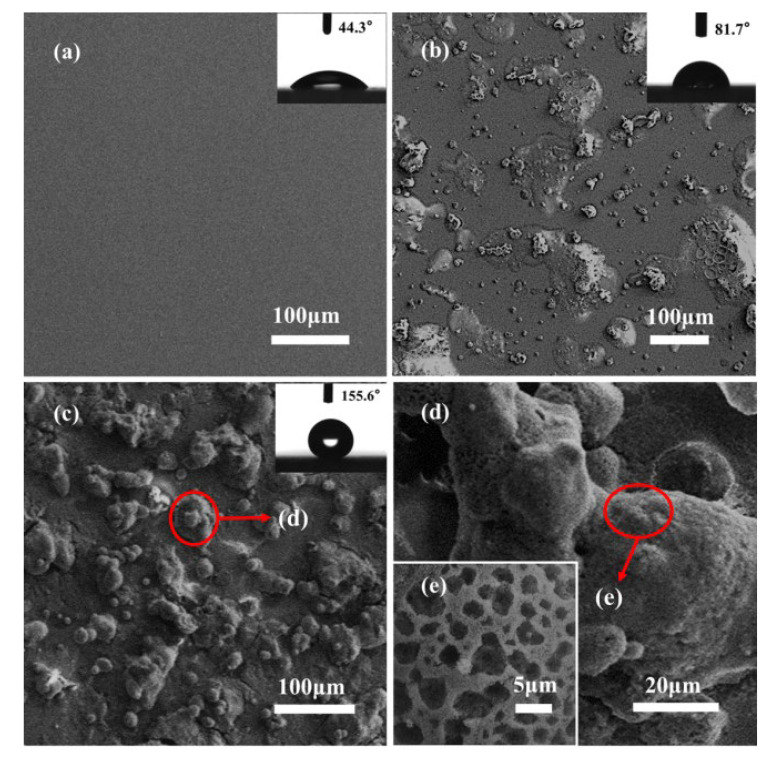
Surface of (**a**) clean glass, (**b**) SBS/C9 coatings, and (**c**–**e**) SBS/C9-SiO_2_ coatings under SEM pictures and CA pictures.

**Figure 5 materials-16-01771-f005:**
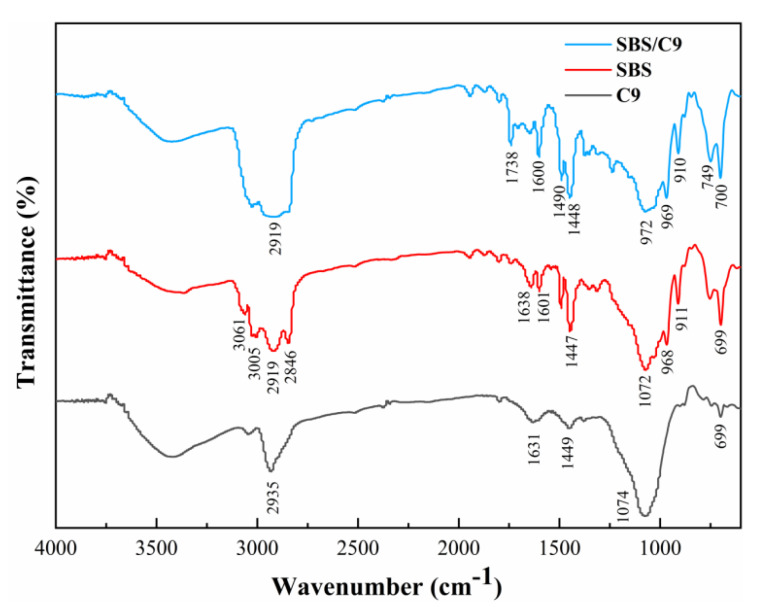
FTIR spectra of SBS, C9 petroleum resin, and SBS/C9.

**Figure 6 materials-16-01771-f006:**
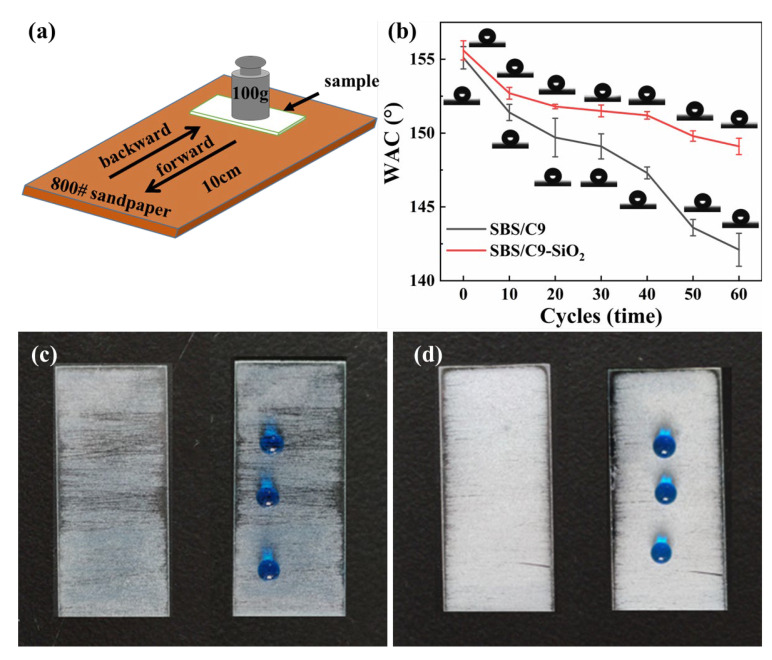
(**a**) Schematic diagram of abrasion resistance test of sandpaper; (**b**) relationship between abrasion times and the CA of the SBS-SiO_2_ coatings and SBS/C9-SiO_2_ coatings; image of the (**c**) SBS-SiO_2_ coatings and (**d**) SBS/C9-SiO_2_ coatings after 60 rubbings.

**Figure 7 materials-16-01771-f007:**
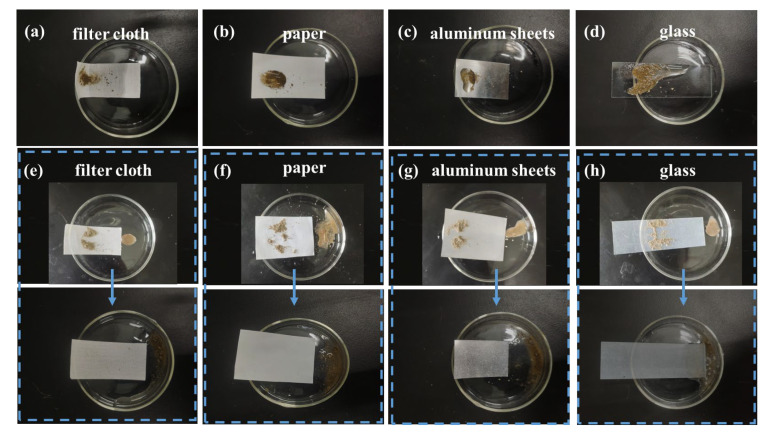
The self-cleaning experiment of SBS/C9-SiO_2_ coatings.

**Figure 8 materials-16-01771-f008:**
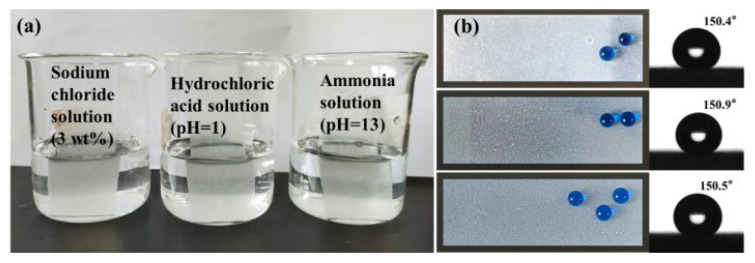
(**a**) The SBS/C9-SiO_2_ coatings were soaked in sodium chloride solution (3 wt.%), hydrochloric acid solution (pH = 1), and ammonia solution (pH = 13); (**b**) glass surface with SBS/C9-SiO_2_ coatings and CA pictures after soaking for 24 h.

**Figure 9 materials-16-01771-f009:**
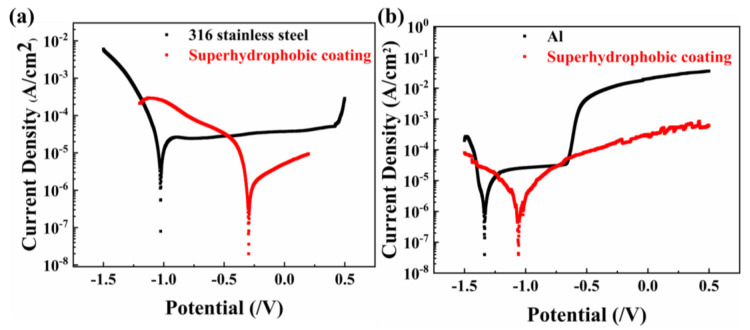
The potentiodynamic polarization curve of (**a**) 316 stainless steel substrate and with SBS/C9-SiO_2_ coatings; and (**b**) Al substrate and with SBS/C9-SiO_2_ coatings.

**Figure 10 materials-16-01771-f010:**
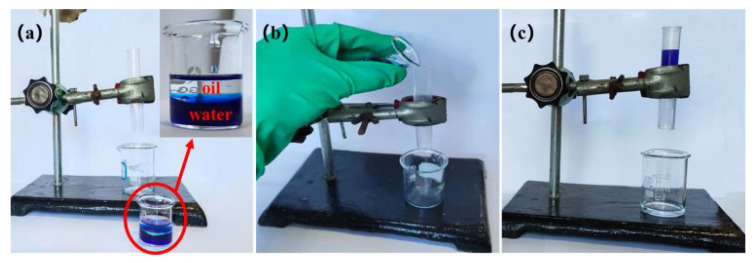
(**a**–**c**) Oil–water separation experiment of the SBS/C9-SiO_2_ coatings.

## Data Availability

Not applicable.

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
