# Peer review of "A Simple and Convenient Method for Preparing Fluorine-Free Durable Superhydrophobic Coatings Suitable for Multiple Substrates"

_materials, 2023, doi:10.3390/ma16051771_

Round 1

Reviewer 1 Report

This paper presents the details of preparing superhydrophobic   coatings on various substrates using   petroleum resin,   styrene-butadiene-styrene and hydrophobic silica.   Mechanical stability, chemical stability, and self cleaning  ability are studied.    The study and results are interesting.  However, the paper requires  a major  revision   suggested below before its further consideration:

English language should be improved throughout for better readability.  There are many typos and linguistic issues, which should be corrected.

 Creating SHP surfaces using low surface energy materials is easy and is well documented. But the challenge is with respect to their mechanical/chemical durability, adherence, and cost-effectiveness.  The above aspects are not properly dealt with in this manuscript and require attention.

The abstract should highlight the problem addressed in this study and its key findings. It should be crisp. Please revise.

Expand abbreviations in the first place where they appear.  

A lot of work is done in this field of SHP materials. The novelty of the present work with respect to those studies should be highlighted.      E.g. Applied Surface Science 585, 152628, 2022; Progress in Organic Coatings 172, 107076, 2022. Authors must discuss the above papers in the revised manuscript and compare the results.

The mechanism of super hydrophilicity must be explained properly. This is very important to support the claims.

Discussion on durability should be improved with convincing data.

Also, the cost comparison and their long-term stability need through studies.

Figure captions should be self-explanatory. 

Discussion of corrosion mechanism should be improved. 

Some Figures require tidying.

Finally, the scientific rigor of the presentation should be improved.

I suggest the authors carefully consider the above aspects during the revision.  

Reviewer 2 Report

This manuscript describes a strategy - spray technique - to obtain functional superhydrophobic coatings, aimed to enhance the adhesion of the coatings to the substrate and to improve the mechanical wear resistance and durability of the surface. The study comprises many methods of investigation, regarding different aspects of the surface chemistry, morphology, wettability, corrosion resistance and chemical stability.  

The manuscript is well organized, documented and the results are well correlated.

After a careful reading of the manuscript, there are some observations and modification to be made.

1. The introduction seems to be too long, please try to shorten, compact it .

2. In materials and methods section, lines 137-138, please revise the type of reference electrode, it was not well defined. A solution does not represent an electrode, something is missing.

3. For testing chemically stability in solutions with different pH, do you think that 24h of immersion in aggressive environments _pH = 1 and pH = 13, are enough?... Have you taken into account a dynamic regime?

4. When referring to the change in corrosion potential, in electrochemistry we used the terms “the corrosion potential was displaced to more electropositive or more electronegative potential”. It should be avoided the terms “rises” or “increases” in order not to create confusions.

5. Please look for lines 283 and 285, there are missing the points that separate 10 to minus (current values) …

6. Why it was chosen as electrolyte solution for electrochemical test, 3.5% NaCl?

Reviewer 3 Report

This manuscript reports the use of petroleum resin as a binder in SBS to increase the mechanical stability of superhydrophobic coatings. The results are very interesting and increase the potential of the formulations containing the described components even if the simultaneous presence of SiO2 nanoparticles appears to be essential to obtain hydrophobicity. However, the manuscript requires major revision and some parts need to clarify.

 In particular,

1)     In the introduction, numerous methods present in the literature are described which cover similar fields of application with formulations similar to the one being studied. The literature review is extensive but could be better organized so that it doesn't look like a simple list of references.

 2)    In the experimental part for FT-IR spectra it is necessary to describe the recording method (ATR? KBr?)

 3)    Results and discussion:

3.4 Surface composition analysis

There are comments in the discussion of FT-IR spectra that should be examined carefully.

The attribution of the peak to 1738 is defined as "a pan-frequency peak": what is meant? Could it instead be attributed to the presence of traces of the ester solvent used for mixing? The spectral range is characteristic for carbonyl groups, for example of esters.

Furthermore, the authors comment in detail on the SBS/C9 spectrum describing broader signals assigned to SBS/C9 while the spectrum of SBS/C9 and SBS shown in the figure are also very similar in bandwidth. The small difference between the two spectra must be justified by the presence of C9 which, however present in a small percentage, does not significantly modify the spectrum of the mixture. Finally, the black spectrum attributed to the C9 blend does not seem to agree with a hydrocarbon composition. Finally, all the spectra show an intense peak around 1070 which is not attributed and which is probably due to the simultaneous presence of SiO2. In conclusion, the authors must carefully check the spectra reported in the figure with the relative attributions integrating the discussion appropriately

Finally, it is wrong to speak of hydrogen bonding because there are no NH or OH or FH groups that can create this type of secondary interaction with N, o or F.

The authors talk about crosslinking but what generates the crosslinking reaction? Perhaps it is more correct to describe a binding effect due to weak secondary interactions (London forces). The components of the formulation are compounds of different complexity but with a hydrocarbon skeleton and also the presence of double bonds would need radical initiators and an increase in temperature to favor cross-linking.

CA and SA measurements

The final average value is reported for CA and SA by carrying out several measurements correctly. However, the standard deviation which must be indicated and which may be significant for the variability of the data is not described.

4)    Conclusions

In the conclusions, the authors should comment in more detail on the results obtained and not simply summarize them. In particular, the authors speak of high hydrophobicity achieved with the mixing of C9 but in reality high values of CA are obtained only in the presence of SiO2 while the role of C9 is above all to increase the stability of the coating. These aspects should be discussed more fully.

 5)    Patents

I think this paragraph was left by mistake

 In conclusion, in my opinion, the authors should review the manuscript and several corrections are necessary for the publication on Materials

Reviewer 4 Report

C9 petroleum resin was mixed with a styrene-butadiene-styrene (SBS) solution and deposited on various substrates. Hydrophobic silica (SiO2) nanoparticles were then deposited onto the C9-SBS binder. Superhydrophobicity was achieved. The main message of the manuscript is that the structured surface showed very good durability according to the results of the tape peeling test and the abrasion test. Other studies were performed to characterise the material and to demonstrate potential applications (oil-water separation, corrosion resistance). The manuscript is straightforward and can be published in Materials. It addresses a major obstacle of superhydrophobic surfaces which is durability. I have only minor concerns.

1. On lines 96-97, the authors wrote: “The mass ratio of SBS to C9 petroleum resin was 1:2 and the hydrophobic SiO2 nanoparticles content was 5 wt.% to produce the SBS/C9-SiO2 coatings with the optimum performance.”

In lines 117-118, the authors wrote: “…SiO2 nanoparticles were dissolved in ethyl acetate and sonicated for 15 minutes to produce 4 wt.% of SiO2 hydrophobic dispersion.”

What is the correct concentration for the SiO2 nanoparticles?

2.The spraying technique which is employed by the authors and praised in lines 43-44 as an excellent tool to induce superhydrophobicity in large surfaces has been recently reviewed in Construction and Building Materials 320 (2022) 126175. In this review paper an alternative method has been suggested: instead of applying a binder onto the substrate prior to the nanoparticle deposition, the NPs are added into the binder sol followed by spraying deposition. The authors may stretch this alternative route to produce superhydrophobic surfaces which is furthermore described in other articles e.g. Coatings 10 (2020) 334.

3. English should be improved.

Round 2

Reviewer 1 Report

The authors have done a good job and the revised version reads well. Therefore, I recommend the revised version for publication. 

Author Response

Thanks very much for your kind work and consideration on the publication of our paper. On behalf of my co-authors, we would like to express our great appreciation to you.

Reviewer 2 Report

The paper was considerably improoved. I agree with its publication. 

Author Response

(The authors gave the same response as above.)

Reviewer 3 Report

The authors have introduced several important reviews trying to accommodate the reviewers' comments.

However, there are still some incorrect points that necessarily require revision.

In particular:

1)    In the introduction, the authors made a comprehensive revision in the introduction, and now the comments on the literature data are better organised. However, some revisions are needed on this part:

Line 77 “dependence” change as “dependent”

Line 96-98 2However, SBS itself is non-polar, its van der Waals force and penetration capabilities are weak, and the bonding effect is not visible, when added to petroleum resin can make it play a greater bonding impact” This sentence needs to be written better

2)    In the experimental part for FT-IR spectra the authors have added some information on how the spectra were recorded but based on their answer in the cover letter some important details are still missing. I suggest to add at line 156-157 “spectra were recorded on films obtained from solutions after solvent evaporation”

3)    Results and discussion:

3.4 Surface composition analysis

Many inaccuracies and errors remain in the comments on the FT-IR spectra but above all I believe it is essential to check the recorded spectra, possibly repeating the analyses. Contaminations must be excluded because the band at around 1070 cm-1 present in all spectra is not attributable to the declared components (or oxygenated products have been used) and is too intense to be attributed to C-H plane bending vibration or C-C single bond backbone vibration.

Furthermore, the attributions for the following bands are wrong:

Lines 238-239: “the peaks at 1490 cm-1 and 1448 cm-1 were are the in-plane and out-of-plane bending vibrations of the methyl C-H bond on the aromatic ring” In that area there are C-H bond deformation for aliphatic structure while C-H of aromatic rings go in the area 900-700. All attributions need to be checked better.

Lines 243-246: The comment on the spectral variation for the mixture SBS/C9 vs SBS spectrum is not justified by the observed spectral differences and it is not justified by the presence of London forces either. As already indicated in the previous review:

“… the authors comment in detail on the SBS/C9 spectrum describing broader signals assigned to SBS/C9 while the spectrum of SBS/C9 and SBS shown in the figure are also very similar in bandwidth. The small difference between the two spectra must be justified by the presence of C9 which, however present in a small percentage, does not significantly modify the spectrum of the mixture.”

 Finally, I reiterate that the black spectrum attributed to the C9 blend does not add up with a hydrocarbon composition.

 In conclusion, in my opinion, the authors should further review the manuscript and several corrections are still needed for publication in Materials
